# STEERING LANGUAGE MODELS FOR THEOREM PROVING

## ABSTRACT

Recent advances in automated theorem proving use Large Language Models (LLMs) to translate informal mathematical statements into formal proofs. However, informal cues are often ambiguous or lack strict logical structure, making it hard for models to interpret them precisely. While existing methods achieve strong performance, little is known about how LLMs internally represent informal cues, or how these influence proof generation. To address this, we explore *activation steering*, an inference-time intervention that identifies linear directions in residual activations associated with informal reasoning traces and adjusts them to improve proof construction without fine-tuning. This mechanism also yields interpretable information about how reasoning is internally encoded in the activation space of LLMs. We test our method for generating formal proofs from already-formalized theorems. Our contributions are twofold: (1) a novel activation-based intervention for guiding proof synthesis in LLMs; and (2) demonstration that this intervention improves performance under two decoding strategies (sampling and best-first search) without any further training.

## 1 INTRODUCTION

Interactive proof assistants such as *Lean* (de Moura et al., 2015), *Isabelle* (Wenzel et al., 2008), and *Rocq* (Barras et al., 1999) provide the infrastructure for formal verification of mathematical proofs and software. They require proofs to be expressed in precise formal languages (Avigad, 2023; Ringer et al., 2019). Neural theorem proving, combining LLMs with proof assistants, has shown promise in automating reasoning (First et al., 2023; Polu & Sutskever, 2020; Polu et al., 2022; Yang et al., 2023; Welleck, 2023). Prior work (Lin et al., 2024; Welleck et al., 2021; 2022) suggests that training model on natural reasoning for proof steps can improve performance, but it remains unclear how such informal language is internally represented or whether it can be reliably leveraged to improve proof generation.

Building on the observation that proofs often interleave natural-language reasoning with formal steps, we show that informal natural language (NL) context induces distinct activation patterns in LLMs. We extract these patterns as steering vectors as in (Panickssery et al., 2024b; Turner et al., 2024; Lucchetti & Guha, 2024), and use them to intervene at inference time. These interventions improve proof generation quality, while preserving model behavior in unrelated aspects. We test our approach across three theorem-proving oriented LLMs - *Lemma* (Azerbayev et al., 2024), *InternLM-2* (Ying et al., 2024), and *InternLM2.5-StepProver* (Wu et al., 2024). On established benchmarks *MiniF2F* and *PutnamBench*, we find that steering via informal language context consistently improves proof quality under multiple decoding strategies.

Our key contributions are threefold: (1) We provide mechanistic insights into how informal mathematical or natural language context impacts internal reasoning in LLMs for theorem proving. (2) We propose a method to extract activation vectors that encode such informal context, and use these vectors to guide proof construction in a structured, grounded way. (3) We demonstrate that activation steering yields consistent improvements in proof success rates on *MiniF2F* and *PutnamBench* benchmarks under both search- and sampling-based decoding.

By focusing on steering in activation space, our approach sheds light on the link between informal mathematical language and formal reasoning steps; without needing to fine-tune model weights. We base this on the hypothesis that many reasoning features are encoded as (approximately) linear

directions in activation space (Mikolov et al., 2013; Elhage et al., 2021; Nanda et al., 2023; Park et al., 2024). Although not all features follow perfect linearity (Engels et al., 2025), this assumption has previously enabled methods like concept erasure and steering to work well (Beaglehole et al., 2025; Zhao et al., 2025a; Shah et al., 2025).

The rest of this paper is organized as follows. In Section 2, we review related work on neural theorem proving, activation steering, and the representation of informal mathematical reasoning in LLMs. Section 3 presents our activation steering method: how steering vectors are constructed from informal mathematical contexts, how they are applied at inference time, and how we select layers and intervention strengths. Section 5 describes our experimental setup, including models, benchmarks (*MiniF2F*, *PutnamBench*), and decoding strategies (sampling vs best-first search). Section 6 reports results: we analyze performance gains, the effect of steering vectors on proof success rates, and ablations experiments. Finally, in Section 7 we discuss insights into the internal reasoning of the model, limitations of our approach, and potential directions for future work.

## 2 RELATED WORK

This section situates our work in three intersecting strands: automatic theorem proving and auto-formalization, representation learning in large language models (LLMs), and mechanistic interpretability via activation interventions.

**Automatic Theorem Proving and Auto-formalization**  Recent progress in automatic theorem proving frames proof search as sequence generation. The GPT-f framework (Polu & Sutskever, 2020) trains a language model to map proof states to tactics, combining it with best-first search to assemble proofs. Extensions explore data augmentation via proof transformations or synthesis (Han et al., 2022; Rotella et al., 2025; Wang & Deng, 2020), improved search strategies (Wang et al., 2023), curriculum training (Polu et al., 2022), and retrieval from proof libraries (Yang et al., 2023). Systems like LLMStep unify these ideas into usable frameworks (Welleck & Saha, 2023).

Auto-formalization complements this by translating informal mathematics (e.g. text, sketches) into formal proofs. Surveys highlight translation techniques (Wu et al., 2022), while Draft-Sketch-Prove shows LLMs prompted with informal sketches outperform formal-only baselines (Jiang et al., 2023). LeanStar (Lin et al., 2024) interleaves informal reasoning with tactic prediction, typically via super-vised fine-tuning on synthetic data. A central open question remains: *How do informal reasoning patterns inform formal proving within a model's internal representations?* Our approach explores this via *steering vectors* in activation space, injecting implicit natural-language "thoughts" at in-ference to guide proof steps without heavy fine-tuning, probing the latent link between informal intuition and formal reasoning.

**Language Model Representation of Concepts**  Our method is motivated by prior work demon-strating that task or feature concepts can be represented as linear directions in the activation space of LLMs when given appropriate contextual examples. Hendel et al. (2023) and Todd et al. (2024) show that a context (e.g. a prompt) can induce a task embedding in some activation subspace. A broader literature examines linear decompositions of features such as truthfulness (Azaria & Mitchell, 2023; Li et al., 2023; Marks & Tegmark, 2024), sentiment (Tigges et al., 2024), harmlessness (Zou et al., 2025; Zheng et al., 2024), sycophancy or alignment steering (Perez et al., 2023; Panickssery et al., 2024a; Sharma et al., 2024), factual knowledge (Gurnee & Tegmark, 2024), and refusal behavior (Arditi et al., 2024). Relatedly, unsupervised methods like sparse autoencoders have been used to extract concept directions in hidden space (Bricken et al., 2023; Huben et al., 2024; Templeton et al., 2024). These works generally share the hypothesis that LLMs encode high-level features or con-cepts as (approximately) linear directions in activation space (Elhage et al., 2021; Mikolov et al., 2013; Nanda et al., 2023; Park et al., 2024). While recent work cautions that not all features may admit clean linear representations (Engels et al., 2025), the linearity assumption has proven effective in practice for concept erasure, model steering, and interpretability (Beaglehole et al., 2025; Zhao et al., 2025a; Shah et al., 2025). Within this framework, we explore whether *generating a natural-language informal explanation can itself be represented as a linear direction, and how injecting this direction at inference improves formal theorem proving performance*.

**Mechanistic Interpretability and Activation Patching**  A rich body of work in mechanistic in-terpretability seeks to locate, analyze, and manipulate internal representations in transformer-based

models. Early studies localized factual knowledge or associative memory to particular neurons or circuits (Meng et al., 2022), and probed hidden layers for high-level features (Li et al., 2024; Dong et al., 2023). The idea of implicit evaluation i.e. measuring latent capability of a model rather than just output behavior, has been developed to complement benchmark-based evaluation (Dong et al., 2023). One particularly relevant tool is activation patching (or residual stream intervention) (Vig et al., 2020; Variengien & Winsor, 2023), wherein one alters specific activations in a layer (often in the residual stream) at inference time to influence model outputs. Recent works have used this to edit factual associations or steer behavior (Zhao et al., 2025b). The residual stream is a high-dimensional accumulator of intermediate features (propagated via skip connections) that each transformer layer refines or adds to; intervening there offers a principled way to steer model behavior (Elhage et al., 2021). In the domain of theorem proving, activation interventions provide a promising lens into *how a model processes and integrates informal guidance with formal reasoning*. In our approach, we generate steering vectors (patches) that, when applied to the residual stream at certain layers, help the model emit an internal "natural-language thought" prior to or alongside tactic prediction. We find that effective steering tends to realign activations such that the output of the model conforms to a structured reasoning format, thereby improving downstream proof search and reducing failure modes.

Building on these insights, we adapt activation interventions to theorem proving by treating informal reasoning as a direction in activation space. We then show how extracting and injecting this direction can guide proof generation. The next section details the architecture, design choices, and evaluation of this steering approach.

# 3 STEERING FOR IMPROVING THEOREM PROVING

We aim to steer theorem-proving models toward using informal reasoning via activation interventions. This section describes (i) how we compute steering vectors, (ii) how we choose layers and apply steering, and (iii) efficiency and practical considerations.

## 3.1 INTUITION AND OVERVIEW

Modern transformer models often encode high-level semantic behavior (e.g. reasoning, style) as approximately linear directions in activation space (residual streams) (Zou et al., 2025; Elhage et al., 2021). We exploit this property: given pairs of prompts that differ only by the presence of natural-language reasoning (but otherwise describe the same proof step), we can estimate a vector in activation space that points in the "informal reasoning" direction. At inference time, adding this vector (appropriately scaled) nudges the model toward producing reasoning-augmented proofs.

In the rest of this section, we detail how we compute these steering vectors, how we pick which layers to intervene on, and how we incorporate them efficiently at inference.

## 3.2 CONSTRUCTING STEERING VECTORS

We adopt a difference-of-means (contrastive) method (Belrose et al., 2023), which has been effective at extracting feature directions across multiple domains (e.g. refusal, truthfulness) (Arditi et al., 2024; Panickssery et al., 2024b; Marks & Tegmark, 2024). Let $\mathcal{D} = \{(p_i, p_i^+)\}_{i=1}^N$ be a dataset of paired prompts, where $p_i$ is a standard formal proof prompt and $p_i^+$ is its version augmented with explicit natural-language reasoning. We feed both prompts into our model $\mathcal{M}$ and extract the residual stream activations at a chosen layer $\ell$. Denote the activation at the final token position by $\mathbf{v}_\ell^p = \mathrm{resid}(\mathcal{M}(p), \ell)$.

Then the steering vector $\mathbf{u}^{(\ell)}$ is computed as:

$$\mathbf{u}^{(\ell)} = \frac{1}{|\mathcal{D}^+|} \sum_{p^+ \in \mathcal{D}^+} \mathbf{v}_\ell^{p^+} - \frac{1}{|\mathcal{D}^-|} \sum_{p^- \in \mathcal{D}^-} \mathbf{v}_\ell^{p^-} \tag{1}$$

Here $\mathcal{D}^+$ and $\mathcal{D}^-$ are the two halves of the prompt-pair dataset (augmented and unaugmented), so the subtraction isolates the direction most correlated with natural-language reasoning while largely cancelling out shared biases or irrelevant activation patterns.

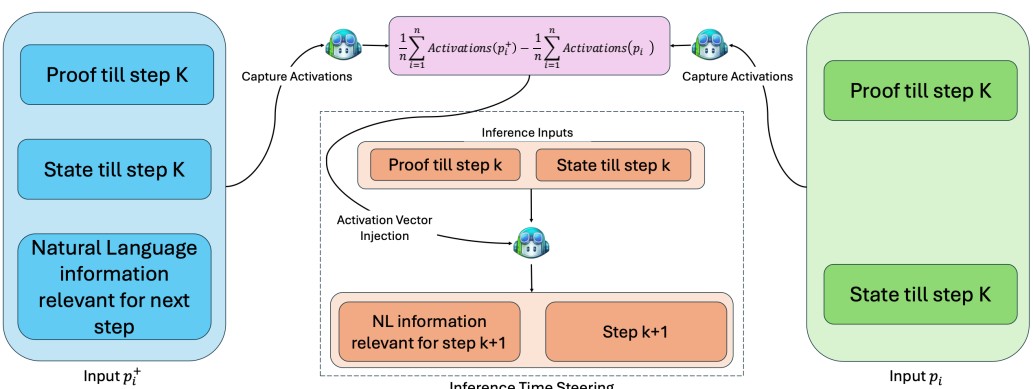

Figure 1: Steering Vectors are computed as difference of means for $p^+$ and $p$

We emphasize that this method requires only forward passes; no gradient-based finetuning or parameter updates; making it computationally lightweight.

### 3.3 LAYER SELECTION AND INTERVENTION STRATEGY

Not all layers are equally responsive to steering, so we first perform an activation analysis to find suitable intervention points. For each layer $\ell$, we measure:

$$\text{sim}(\ell) = \frac{1}{|\mathcal{D}|} \sum_{(p,p^+)\in\mathcal{D}} \cos(\mathbf{v}_\ell^p, \mathbf{v}_\ell^{p^+}) \tag{2}$$

Empirically, we observe that in early layers, $\text{sim}(\ell)$ is close to 1 (i.e. little divergence), but in deeper layers it drops and exhibits local minima ("valleys") as seen in Figure 2.

We find that in early layers, activations remain highly similar between $p$ and $p^+$, but as we go deeper, the cosine similarity drops and exhibits local minima. Intuitively, these are the layers where the representations diverge most when informal reasoning is introduced (see Figure 2). Thus, these "valley" layers are promising candidates for steering. These valleys likely correspond to points where the internal representation of the model is most sensitive to reasoning-specific perturbations.

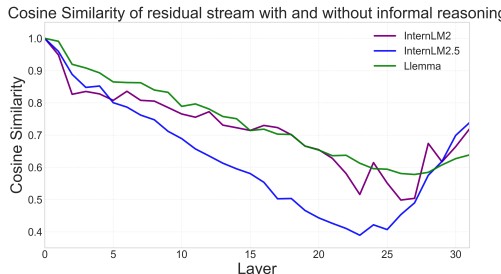

Figure 2: Cosine similarity across model's residual stream activations for each layer.

We select intervention layers where (a) cosine similarity dips significantly, and (b) representation is still semantically stable. At inference, we inject the steering vector into those layers:

$$\mathbf{v}'_\ell = \mathbf{v}_\ell + \alpha \cdot \mathbf{u}^{(\ell)} \tag{3}$$

We treat $\alpha$ as a hyperparameter and specify that it will be tuned on held-out validation data. The method instantiates a sweep to find the optimal balance between influencing reasoning and maintaining validity. We place constraints (e.g. an upper bound on $\alpha$) to avoid destabilizing proofs. We present the complete pseudocode of our approach in Algorithm 1.

---

**Algorithm 1** Activation Steering for Theorem Proving

---

**Require:** Contrastive prompt pairs $D = \{(p_i, p_i^+)\}_{i=1}^N$ where $p_i$ are standard prompts and $p_i^+$ include natural language reasoning
**Require:** Language model $M$ with $L$ layers
**Require:** Test theorem $T$ to prove
**Ensure:** Enhanced proof generation
 1: **Offline: Create Steering Vectors**
 2: **for** each layer $\ell \in \{1, \ldots, L\}$ **do**
 3:     $\bar{v}^+ \leftarrow \frac{1}{N} \sum_{i=1}^N \text{resid}(M(p_i^+), \ell)$ {Mean activation for augmented prompts}
 4:     $\bar{v}^- \leftarrow \frac{1}{N} \sum_{i=1}^N \text{resid}(M(p_i), \ell)$ {Mean activation for standard prompts}
 5:     $u^{(\ell)} \leftarrow \bar{v}^+ - \bar{v}^-$ {Steering vector at layer $\ell$}
 6: **end for**
 7: $\mathcal{L} \leftarrow \text{SELECTLAYERS}(\{u^{(\ell)}\}_{\ell=1}^L, D)$ {Identify optimal intervention layers}
 8: **Online: Apply Steering During Inference**
 9: Initialize proof context with theorem $T$
10: **while** proof incomplete and within computational budget **do**
11:     Perform forward pass through model
12:     **for** each intervention layer $\ell \in \mathcal{L}$ **do**
13:         $v'_\ell \leftarrow v_\ell + \alpha \cdot u^{(\ell)}$ {Inject steering vector}
14:     **end for**
15:     Generate next proof step using modified activations
16:     Validate and apply proof step if correct
17: **end while**
18: **return** Generated proof or failure

---

# 4 EXPERIMENTAL SETUP

## 4.1 MODELS

We conduct experiments on three open-source 7B-parameter language models optimized for mathematical reasoning:

- **Llemma-7B** (Azerbayev et al., 2024): Built on Code Llama architecture, pre-trained on mathematical texts and formal mathematics corpora.

- **InternLM2-7B** (Ying et al., 2024): LLaMA-based architecture with extended training on mathematical problem-solving.

- **InternLM2.5-StepProver** (Wu et al., 2024): Enhanced variant with expert iteration on large-scale Lean problems.

All models share transformer architectures with consistent residual stream dimensionality, facilitating uniform steering vector extraction.

## 4.2 DATA AND VECTOR CONSTRUCTION

We construct steering vectors from the Lean-STaR dataset(Lin et al., 2024), randomly sampling 10,000 theorem-proof pairs. Each datapoint $(p, p^+)$ consists of a formal proof step $p$ and its augmented version $p^+$ containing explicit natural language reasoning.

For robustness, we generate model responses $r_p$ and $r_{p^+}$ for each prompt pair and retain only instances where both responses are valid proof steps with $r_p \neq r_{p^+}$. This filtering yields approximately 7,400 high-quality contrastive pairs.

Vector construction requires only forward passes through the model, consuming $\sim 1$ GPU-hour on NVIDIA A100, a 60× reduction compared to LoRA fine-tuning while achieving superior performance gains.

### 4.3 Chosen Layers and $\alpha$

Based on activation similarity analysis (Section 3.3), we apply steering vectors to Layers 22, 25 in **InternLM2-7B** and Layers 24, 25 in **Llemma-7B**. These correspond to layers where the model's representation diverges most from baseline while maintaining semantic coherence. We empirically determine $\alpha = 0.8$ through validation experiments.

### 4.4 Evaluation Benchmarks

**miniF2F**   Our primary evaluation utilizes miniF2F (Zheng et al., 2022), a standardized benchmark comprising 244 theorems drawn from mathematical competitions (AMC, AIME, IMO). These problems span algebra, number theory, geometry, and combinatorics with varying proof complexities. We employ two proof search strategies:

- **Best-first search**: Expansion budget $N \in \{50, 600\}$, tactic sampling width $S = 32$, selecting states by cumulative log-probability
- **Parallel sampling**: $K$ independent proof attempts to accommodate probability shifts from natural language injection

**PutnamBench**   We additionally evaluate on PutnamBench (Tsoukalas et al., 2024), containing problems from the William Lowell Putnam Mathematical Competition. Following standard evaluation protocol, we test on both the Lean subset (657 problems) and Rocq subset (412 problems) using the benchmark's default search parameters: $N = 600$ expansion budget with $S = 32$ tactic width for best-first search, and parallel sampling with $K = 2$ attempts. This benchmark features more challenging problems with longer average proof lengths, providing a stringent test of steering effectiveness on complex mathematical reasoning.

**Evaluation Metrics.**   Following established practice in neural theorem proving, we report pass rates (percentage of theorems successfully proved within computational budget) as our primary metric. For detailed analysis, we additionally examine proof characteristics including average length, tactic distribution, and the frequency of intermediate lemma usage (via the `have` tactic) to understand how steering affects proof structure and strategy.

## 5 Results and Analysis

Our experiments were designed to address the following research questions:

1. Does activation steering improve theorem-proving performance of existing models?
2. Which layers contribute most effectively when steered?
3. Does steering enhance proof structure and search efficiency?
4. Are improvements consistent across different search budgets?
5. How effective and efficient are steering vectors compared to LoRA fine-tuning?
6. Do steering vectors generalize across provers?

### 5.1 Q1: Impact of Activation Steering

Activation steering consistently improves model performance across all tested configurations, enabling the discovery of proofs not derivable by the base models alone.

**MiniF2F.** As shown in Table 1, steering improves performance across all models on the *miniF2F-Test* benchmark (Zheng et al., 2022). We adopt the sampling decoding strategy from LeanStar (Lin et al., 2024), which mitigates variance from natural-language generation and provides more effective node selection than standard Best-First Tree Search (see Appendix for Best-First results).

Steering introduces structured natural-language comments that enhance mathematical reasoning and proof generation. Importantly, we find that steering enables a significant number of new proofs beyond the base model's reach, suggesting that steering vectors guide the model toward otherwise

| Model | MiniF2F |
|---|---|
| Llemma 7B | 26.6% |
| InternLM2-7B | 28.7% |
| InternLM2.5-Step Prover | 48.2% |
| LLEMMA-7B + Steering | 28.1% |
| InternLM2-7B + Steering | 32.4% |
| InternLM2.5-StepProver + Steering | 66.4% |

Table 1: Performance on MiniF2F (sampling decoding, 50×32×1).

| Model | PutnamBench |
|---|---|
| InternLM2 7B | 4 (0.6%) |
| InternLM2.5-StepProver | 6 (0.9%) |
| InternLM2 7B + Steering | 4 (0.6%) |
| InternLM2.5-StepProver + Steering | 7 (1.1%) |

Table 2: Performance on PutnamBench (Lean). Results reported out of 657 attempts.

inaccessible solution paths. Interestingly, models occasionally succeed despite incorrect informal reasoning, implying that steering exerts influence beyond surface-level natural language.

While steering increases successful proofs, it also introduces more failure cases. We hypothesize this stems from biases in the data used to construct steering vectors, which encode inductive priors that align well with some theorem classes but misalign with others. Despite this trade-off, the net effect is strongly positive, establishing activation steering as a lightweight, parameter-free alternative to fine-tuning.

**PutnamBench.** On PutnamBench (Table 2), steering improves success rates for both InternLM2 and InternLM2.5 on Lean, and achieves non-trivial gains on Rocq problems (details in Section A.3). These results highlight steering's ability to support reasoning in more complex, multi-step proofs.

## 5.2 Q2: LAYER CHOICE

We evaluate layer sensitivity following the strategy described in Section 3.3. Steering vectors are patched layer-wise to analyze their influence. Figure 4 and Table 3 show that later layers exert stronger influence on theorem-proving performance.

| Selected Layers | Pass Rate (%) |
|---|---|
| 25–30 (Late) | 51.6 |
| 14–24 (Middle) | 50.8 |
| 5–13 (Early) | 49.5 |

Figure 3: Pass rates at $2 \times 32 \times 600$ for InternLM2.5-StepProver on miniF2F.

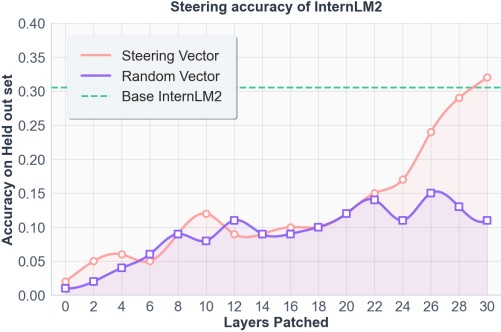

Figure 4: Layer-wise ablation results.

These findings suggest that steering vectors primarily affect later-stage reasoning circuits. To ensure robustness, we also compare against random steering vectors, confirming that observed improvements arise from semantically meaningful directions rather than incidental noise.

## 5.3 Q3: PROOF STRUCTURE AND SEARCH EFFICIENCY

Table 3 shows that steering primarily benefits shorter proofs (<5 steps), while gains diminish for longer ones (>15 steps). Qualitative analysis suggests that informal reasoning introduces noisy intermediate steps for long proofs, reducing effectiveness. Interestingly, steering sometimes shortens proof length by guiding the model toward alternate proof strategies.

| Proof Length | Without Steering | With Steering (total) |
|---|---|---|
| <5 | 76 | 83 (94) |
| 6–10 | 16 | 13 (19) |
| 10–15 | 10 | 11 (20) |
| 15–30 | 15 | 13 (18) |
| >30 | 11 | 8 (11) |

Table 3: Proof length analysis for InternLM2.5-StepProver on miniF2F.

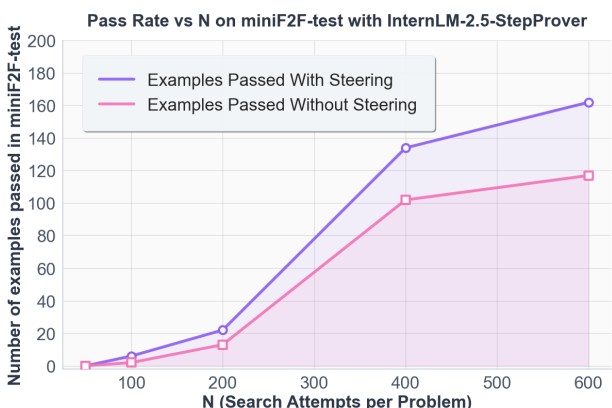

Figure 5: Pass rates on *miniF2F* with varying search budgets.

While new proofs increase substantially, so do failure cases, likely reflecting inductive biases encoded in steering vectors. Characterizing these biases will be crucial for extending steering to broader proof distributions.

### 5.4 Q4: SEARCH BUDGET

Figure 5 shows that steering yields consistent improvements across all search budgets $N$. Gains grow with larger budgets: at $N = 100$, steering triples performance (6 vs. 2 proofs), and at $N = 600$, steering delivers a 38% improvement (162 vs. 117 proofs). These results suggest that steering vectors not only boost baseline performance but also scale effectively with additional computational resources by guiding the search toward more productive proof trajectories.

### 5.5 Q5: EFFECTIVENESS VS. LORA FINE-TUNING

We compare steering against LoRA fine-tuning (Hu et al., 2021) on InternLM2. LoRA models were trained on $p^+$ prompts with (rank=8, $\alpha_{\text{LoRA}}$=16) and (rank=32, $\alpha_{\text{LoRA}}$=64). Performance is reported on **miniF2F** after each epoch (Figure 6).

LoRA with higher rank outperforms steering, but lower-rank adaptations underperform, highlighting their limited representational capacity. In contrast, steering achieves competitive performance immediately, without additional training or parameters. This demonstrates that activation steering offers a highly parameter-efficient alternative, complementing but not replacing fine-tuning.

### 5.6 Q6: GENERALIZATION ACROSS PROVERS

We evaluate if the steering vector is generalizable across different provers. In particular we use Putnambench (Tsoukalas et al., 2024) for evaluating cross lingual transfer for steering vectors. A particularly compelling finding emerges from our cross-system evaluation: despite the steering vectors being derived from Lean-based datasets, the augmented model successfully proves one Rocq problem an improvement over the base model's zero success rate. Specifically, InternLM2.5 with steering successfully constructs a proof for *Putnam 1988 B1* in Roc1, while the same model fails

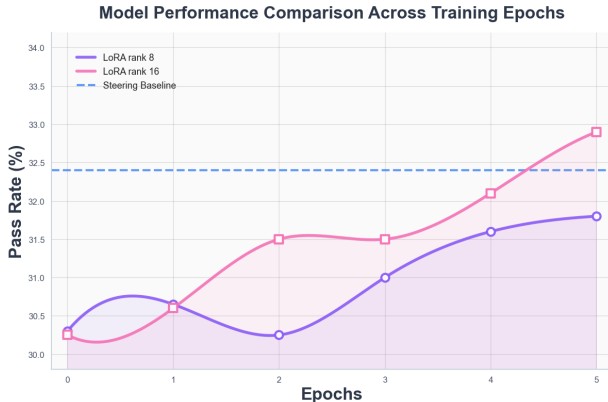

Figure 6: Comparison of steering and LoRA fine-tuning.

to complete the corresponding proof in Lean. In Appendix A.3 we showcase the proof written by InternLM2.5-StepProver.

This preliminary cross-system success suggests that steering vectors may capture reasoning patterns transferable across systems. The transferability suggests that our approach enables model to capture deeper mathematical concepts that transcend the specific formal language, encoding universal reasoning strategies that benefit theorem proving across different proof assistants. This finding has important implications for developing general-purpose mathematical reasoning systems, indicating that steering vectors trained on one formal system may provide value across the broader ecosystem of interactive theorem provers.

## 6    CONCLUSION

Our work presents a deep analysis of the underlying mechanisms that drive natural language thoughts in Large Language Models for formal theorem proving. By isolating and manipulating specific activation directions associated with natural language "thoughts", we have demonstrated a method to enhance LLMs' capabilities in mathematical theorem proving without requiring costly fine-tuning. Our experiments showcase how language models comprehend natural language very differently compared to formal proof steps and theorems. We explore the use of activation vectors to represent the task of producing NL information (or thought) with the proof step. And it performs consistently better than base models trained for theorem proving. We provide a closer look into how language models represent theorems in activation space to generate proofs. The use of steering vectors to isolate layers responsible for formal reasoning shows the promise in this approach. We believe this provides insight into the challenges LLMs face on Olympiad-level problems and how activation steering may mitigate them.

## 7    LIMITATIONS AND FUTURE WORK

This work investigates the use of activation steering to interpret and enhance the performance of language models on theorem-proving tasks. While our explorations are thorough and yield consistent empirical results, there are several limitations worth noting.

First, our study focuses solely on incorporating natural language information to guide the theorem-proving process. However, we do not systematically evaluate the relevance or factual correctness of the natural language inputs with respect to the underlying proof obligations. In certain cases, the model may generate correct tactics even when the provided natural language guidance is partially incorrect or misleading. Understanding this phenomenon requires a deeper analysis of the relationship between instruction quality and proof validity, which we leave for future work.

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
