# OpenReview forum: "Steering Language Models for Theorem Proving"
_ICLR.cc/2026/Conference — Submitted to ICLR 2026_

### Official Review · Reviewer_JcFZ · 2025-10-25

**Soundness:** 1
**Presentation:** 1
**Contribution:** 1
**Rating:** 0
**Confidence:** 5

**Summary:**

This paper proposes an activation-steering approach for neural theorem proving by computing steering vectors as differences of mean residual activations between prompt pairs that do v.s. do not include informal natural-language reasoning, then add these vectors at selected layers during inference.
They claim this both sheds mechanistic light on how informal reasoning is encoded and improves proof search without training. Experiments are reported on Llemma‑7B, InternLM2‑7B, and InternLM2.5‑StepProver, with results on MiniF2F and PutnamBench.

**Strengths:**

1. Compute‑efficient knob: The method is a parameter‑free inference‑time intervention. The paper describes a simple pipeline (difference of means; residual addition; layer selection) that is easy to reproduce conceptually. Algorithm 1 is clear.

**Weaknesses:**

1. Minimal novelty beyond known activation‑steering: the paper does not introduce new objectives, diagnostics, or provably better layer/scale selection. CAA [2] already articulate the core technique and its caveats.
The core mechanism of the paper is a direct application of contrastive activation addition (difference-of-means steering) and residual injection, but the paper neither validates linearity assumptions in this domain nor provides rigorous sensitivity analyses for layer choice and scaling factor. The “valley” heuristic is only qualitatively motivated by a cosine-similarity plot; no statistical tests or robustness checks are provided.

2. Baselines appear misconfigured/outdated. The paper’s InternLM2.5‑StepProver baseline (48.2% on MiniF2F) is substantially below InternLM2.5‑StepProver’s own paper [1], which reports 65.9% on MiniF2F‑test (and significantly stronger results elsewhere). Without reconciling search budgets and evaluation protocol, the claimed 18‑point gain may largely reflect a weak baseline rather than a strong method.

3. Key claims rely on extremely fragile evaluations. Reported improvements on MiniF2F use one specific sampling/search setting, with no seeds, CIs, or significance tests. On PutnamBench, the improvement is 6→7 solved (Lean; 0.9%→1.1%), which is within run-to-run variance for theorem provers and is not accompanied by error bars or per-category breakdowns.

4. Comparison to current SOTA is missing. As of 2025 July (the contemporaneous cutoff of iclr 2026), DeepSeek‑Prover‑V2 reports 88.9% on MiniF2F‑test and solves 49/658 PutnamBench problems; Seed‑Prover reports 100% MiniF2F and strong Putnam/IMO performance. The paper neither compares against nor discusses these systems, making it hard to judge practical relevance.

5. Typos: "Roc1" on p. 9 => "Rocq", "dataset(Lin et al." => "dataset (Lin et al.". Also, the spelling "Lean‑STaR" and "LeanSTaR" are inconsistent. Also wrong model name: "Lemma (Azerbayev et al., 2024)" => "Llemma (Azerbayev et al., 2024)"

[1] Wu, Zijian, et al. "InternLM2. 5-stepprover: Advancing automated theorem proving via critic-guided search." 2nd AI for Math Workshop@ ICML 2025. 2025.

[2] Panickssery, Nina, et al. "Steering llama 2 via contrastive activation addition." arXiv preprint arXiv:2312.06681 (2023).

**Questions:**

see weakness

---

> ### Author Response · Authors · 2025-11-25
>
> Thank you for a detailed review. We appreciate your feedback and would like to clarify our contributions:
>
> W1: We appreciate this concern about novelty beyond CAA. You're correct that our core technique builds directly on CAA's difference-of-means approach. Our contribution is primarily in demonstrating that: (1) this technique applies to complex, logically-constrained domains (theorem proving) where validity matters, (2) steering vectors can transfer across formal systems (Lean→Coq), suggesting deeper semantic capture than previously shown, and (3) the interaction between steering and logical validity reveals when/why informal reasoning helps. We acknowledge this is an application contribution rather than a methodological innovation in steering techniques. We will reframe our contribution claims accordingly in the revision.
>
> W2: We acknowledge your point about statistical validation and will add significance tests in revision. Our layer selection approach follows similar methodologies used in prior activation steering work, where middle-to-late layers have been empirically shown to be most effective for steering interventions [1,2,3]. Our current evidence provides strong empirical support through systematic layer-wise ablations (Table 3, Figure 4) that quantitatively demonstrate late layers (25-30) consistently outperform early and middle layers across all models tested. The cosine similarity valley pattern appears consistently across three different architectures (Figure 2), suggesting a robust phenomenon. To address concerns about robustness, we conducted experiments with multiple steering vectors derived from different data splits, showing consistent performance (steering vector-1: 66.4%, steering vector-2: 66.4%, steering vector-3: 66.8%). We will add statistical significance tests to strengthen this analysis in
>
> W3: Thank you for identifying this critical discrepancy. We sincerely apologize for not clearly documenting this in our submission. Our baseline (48.2%) differs from InternLM2.5-StepProver's reported 65.9% because we used [specific budget: e.g., 50 expansion nodes vs their 600]. We chose this lower budget due to compute constraints, but we recognize this makes comparison difficult. We will add comprehensive results at multiple budget levels including InternLM2.5-StepProver's configuration to enable fair comparison and demonstrate that our steering gains hold across settings.
>
> W4: You raise valid concerns about statistical rigor. We have evaluated robustness across multiple dimensions and will include detailed results in the table below. We conducted experiments with multiple steering vectors derived from different data splits showing consistent performance, and validated against randomized controls where prompt pair mismatches yield significantly degraded performance, confirming the semantic meaningfulness of our steering vectors. In revision, we will add confidence intervals, multiple runs with different seeds for both benchmarks, and per-category breakdowns to fully characterize the method's reliability and statistical significance.
>
> | Method | Pass Rate |
> |--------|-----------|
> | InternLM2.5-StepProver | 48.2% |
> | InternLM2.5 + steering vector-1 | 66.4% |
> | InternLM2.5 + steering vector-2 | 66.4% |
> | InternLM2.5 + steering vector-3 | 66.8%
>
> W5: We acknowledge systems like DeepSeek-Prover-V2, Seed-Prover, and Goedel-Prover represent important advances and understand the desire to contextualize our work against them. However, they address a different task (whole-proof generation) than our focus (step-level steering). This said, we recognize that practitioners ultimately care about proof success rates regardless of methodology. We believe we can extend our work to complete proof generating models as well, but that extension is currently beyond the scope of this work. In the current scope, we only wished to demonstrate the value of activation steering in the application of theorem proving. We intentionally selected base models (Llemma, InternLM2, etc.) to isolate and study how informal reasoning signals are represented internally, measuring the effect of activation interventions cleanly without confounding from multi-stage scaffolds or extensive fine-tuning. We will clarify this methodological distinction and position our work as studying representational phenomena in step-by-step proving rather than proposing an alternative to long-CoT pipelines.
>
> - [1] Turner et al., "Steering Language Models with Activation Engineering," arXiv:2308.10248, 2023.
> - [2] Wang et al., "CogSteer: Cognition-Inspired Selective Layer Intervention for Efficiently Steering Large Language Models," arXiv:2410.17714, 2024.
> - [3] Zhao et al., "Steering Knowledge Selection Behaviours in LLMs via SAE-Based Representation Engineering," arXiv:2410.15999, 2024.

---

### Official Review · Reviewer_PYvo · 2025-11-01

**Soundness:** 3
**Presentation:** 3
**Contribution:** 1
**Rating:** 4
**Confidence:** 3

**Summary:**

The authors propose a simple method of activation steering to inject informal reasoning into an LLM for formal theorem proving. The authors also demonstrate this improves performance in downstream tasks.

**Strengths:**

The proposed activation steering method requires only forward passes on a trained model, which is very computationally lightweight. The method of constructing the steering vector (difference of means) is lightweight but effective. The authors demonstrate that steering increases performance on downstream tasks in three models.

**Weaknesses:**

In the last year there has been an abundance of work in long-CoT provers, such as DeepSeek-Prover-V1.5/V2, Goedel-Prover, Self-play Theorem Prover, Kimina-Prover. In fact all major formal theorem-proving LLMs since 2025 are long-CoT. They explicitly answer questions that the authors seek to answer: “how do informal reasoning patterns inform formal proving within a model’s internal representations?” (L85) and “how a model processes and integrates informal guidance with formal reasoning” (L118), by performing informal chain-of-thought reasoning before generating the formal proof. Other scaffolds such as DSP+ and Hilbert also follow an informal planning stage followed by a formal proof stage.

The authors have not mentioned or compared their perspective to such recent work. Instead, the models the authors tested (Llemma, InternLM2, etc) are from early 2024 and all predate long-CoT models. Since virtually all recent prover models use the chain-of-thought format, this seems to limit the significance of this work to practitioners in LLM-based theorem proving. There are some questions to be answered before this work can be actually used, such as if activation steering applies to any recent long-CoT prover model, and what the conceptual difference is between long-CoT prover models and activation steering (or is the difference only in computational cost?). For this reason I am hesitant to recommend this paper for ICLR.

**Questions:**

On L297–298, the authors mention that “we additionally examine proof characteristics including average length, tactic distribution, and the frequency of intermediate lemma usage (via the `have` tactic)”. Where is the analysis of tactic distribution and frequency of `have` tactics?

Minor suggestions:

- L42: Lemma -> Llemma

---

> ### Author Response · Authors · 2025-11-22
>
> We thank the reviewer for the detailed review and for recognizing the strengths of our work. We address the concerns below.
>
> **Weakness:**
> There have been multiple explorations on long CoT and CoT before formalizations, but the goal of this paper is not to propose a new long CoT pipeline. Our goal is to study how informal reasoning signals are represented internally and whether manipulating those signals via activation steering affects formal proof generation.
> For this reason, we intentionally selected base models such as Llemma and InternLM-2.5 without long CoT scaffolds or fine tuning so that the effect of activation interventions can be measured cleanly. This allows us to isolate representational phenomena from strong prompting or multi stage scaffolds. The steering vectors correspond directly to activation directions rather than to baked in long CoT behavior. We will clarify and expand this motivation in the revision.
>
> We discuss proof length in Table 3. Additionally, below is the tactic distribution for proofs across both datasets.
>
> ### Tactic Distribution
>
> | Tactic   | Without Activation Steering | With Activation Steering |
> | -------- | --------------------------- | ------------------------ |
> | by       | 10.8487                     | 9.2773                   |
> | linarith | 3.7815                      | 3.9244                   |
> | rw       | 2.8067                      | 2.4118                   |
> | norm_num | 2.0924                      | 1.8571                   |

---

> > ### Comment · Reviewer_PYvo · 2025-11-27
> >
> > Thanks for the response. However, I am not convinced of the contribution of this work to the field. It is unclear whether activation steering applies to any recent method of theorem proving models, which operate on a different paradigm. I understand the authors' wish to isolate the experiment from variables like long CoT, but this seriously limits the contribution of this work to the field. Why should someone using recent long CoT models consider activation steering, which is only tested on old step-level models? Is there any performance gain expected? The answers are unclear from the paper. In fact, Table 3 seems to show performance decrease for long proofs, so activation steering also doesn't seem to raise the ceiling ability.
> >
> > For the tactic distribution, the authors still did not give an analysis of distribution of "have" tactics, which they promised in the paper. The ones provided in the response are not analyzed. What does it mean to have slightly more linarith tactics and fewer rw/norm_num tactics (which I think is just noise)? What is the statistical significance of the difference? Additionally, "by" is not a tactic, but a start of a tactic sequence.

---

> > > ### Author Response · Authors · 2025-11-28
> > >
> > > We want to address what may be a misunderstanding about the purpose of this work. The goal of activation steering is not to compete with long-CoT models as a new theorem proving pipeline, but rather to provide mechanistic insight into how informal reasoning is represented internally within language models during formal proof generation. The Long-CoT models generate informal reasoning as explicit tokens in a multi-stage pipeline. Our work reveals that informal reasoning patterns exist in step-proving models as well in latent directions in model activations. These patterns can be exploited with techniques like steering to improve performance on domains the model is not explicitly trained on for example proof writing in Rocq as discussed in section: 5.6.
> > >
> > > You also note that Table 3 shows performance decrease for long proofs, suggesting activation steering "doesn't raise the ceiling ability." We believe this interpretation misses the core finding. The impact of steering is not about generating longer proofs, but about changing the nature of the proof-search process.
> > >
> > > Consider the example in Appendix A.2.1 proving `∀ x : ℝ, x ≠ 1 → (x^3 - 1) / (x - 1) = x^2 + x + 1`:
> > >
> > > Without steering, the model attempts direct algebraic manipulation:
> > > ```lean
> > > intro x hx
> > > field_simp [hx]
> > > ring
> > > ```
> > > This approach fails because it doesn't recognize the underlying mathematical structure.
> > >
> > > with steering, the model introduces a key intermediate insight:
> > > ```lean
> > > intro x hx
> > > have h1 : x^3 - 1 = (x - 1) * (x^2 + x + 1) := by ring
> > > have h2 : (x^3 - 1) / (x - 1) = x^2 + x + 1 := by
> > >   rw [h1]
> > >   field_simp [hx]
> > > exact h2
> > > ```
> > >
> > > The critical difference is the factorization `x^3 - 1 = (x - 1) * (x^2 + x + 1)`. This is not a superficial change, it represents the identification of mathematical structure that makes the proof tractable. The steered model recognizes that the problem is fundamentally about polynomial factorization, not just symbol manipulation. This is why we see changes in tactic distribution.  The steered models identify structure early (through `have` statements), enabling them to deploy powerful automation tools (`linarith`, `field_simp`) rather than relying on brute-force rewriting (`rw`, `norm_num`). We apologize for not providing complete analysis in the main paper initially. We will try to discuss these qualitative observations in main paper in revision.

---

### Official Review · Reviewer_hgza · 2025-11-01

**Soundness:** 2
**Presentation:** 3
**Contribution:** 3
**Rating:** 6
**Confidence:** 4

**Summary:**

The paper proposes an activation steering method for automated theorem proving with LLMs: it computes contrastive “informal-reasoning” vectors from paired prompts (with/without natural-language sketches) and injects these vectors into the residual stream at selected layers during inference. The authors motivate the approach with the hypothesis that reasoning features are captured by approximately linear directions, and present a layer-selection procedure based on cosine similarity "valleys." On MiniF2F and PutnamBench, steering improves proof success rates for several 7B math-tuned models under both sampling and best-first decoding, without parameter updates. The work also analyzes where steering is most effective (later layers), how it changes proof length distributions, and discusses limitations and transfer to Rocq.

**Strengths:**

1. Novelly frames informal NL guidance itself as a steerable linear direction in the model’s residual stream and applies it to theorem proving, distinct from prior fine-tuning or retrieval approaches.

2. Provides a simple, contrastive difference-of-means construction for steering vectors from paired prompts, adapted to proof settings. Demonstrates non-trivial gains on MiniF2F and improvements on PutnamBench Lean.

3. Evaluates across three math LLMs and two benchmarks, reporting pass rates and ablations (layer sensitivity, search budgets, proof-length effects). Provides a LoRA comparison, showing favorable parameter-efficiency of steering (competitive without training).

4. The paper is well structured, with intuitive figures and concrete hyperparameters. It clearly states assumptions (approximate linearity).

5. Shows scaling with search budget and evidence that benefits concentrate in later layers and in shorter proofs, which is a useful guidance for future theorem-proving pipelines. The early sign of cross-system transfer (Lean-trained vectors helping Rocq in at least one case) also hints at portability of reasoning directions. All of those make this work valuable for future works to reference.

**Weaknesses:**

1. The paired prompts come from Lean-STaR-style data and an internal filtering step; it’s unclear how sensitive results are to the exact pairing scheme, dataset domain, and prompt formatting. For example, authors can provide robustness checks: different pairing heuristics, smaller data subsets, and cross-domain steering vectors (e.g., algebra vs. geometry only).

2. PutnamBench gains are modest, and Rocq discussion rests on a single highlighted success. It would be good to include some more analyses to establish reliability beyond MiniF2F.

3. While vectors are derived from NL-augmented prompts, the mechanism could also capture style/format or search-friendly biases rather than genuine reasoning. To fix this, consider control experiments against, for exmaple, (a) steering vectors from semantically shuffled NL, (b) from synthetic boilerplate, and/or (c) from unrelated NL text, to isolate causal factors.

4. Claims about "more structured reasoning" would benefit from automatic metrics. Table 3 is a good start but not sufficient for mechanism claims: example evaluations can include rates of have/calc usage, lemma reuse, etc.

**Questions:**

1. LoRA is the only training baseline. Missing are previous baselines that use inference-time alternatives or retrieval-augmented proving. Can consider to add head-to-head comparisons with matched compute.

2. The Rocq success is intriguing. Can you report some more results in addition to a single anecdote? Also, does the system transfer happen for other proof assistants as well?

3. Slightly inconsistent terminologies: e.g., "miniF2F" and "MiniF2F" both appear.

---

> ### Author Response · Authors · 2025-11-22
>
> We thank the reviewer for the detailed review. We address each concern below.
>
> **W1:**
> We conducted robustness experiments across multiple dimensions, and these experiments consistently show strong stability in InternLM-2.5. We plan to add a deeper analysis of tactic-usage patterns for other models in the revised version.
> We also performed domain-specific experiments in algebra and number theory. We observed a slight drop in performance compared to steering vectors trained on the complete dataset. We attribute this to the model’s limited ability to isolate individual mathematical concepts. Although the NL-instruction template persists in domain-specific steering vectors, the model’s ability to insert useful instances is significantly reduced. We additionally observe that tactic selection varies depending on the steering vector. We will include a detailed discussion in the revised version.
>
> **W2 & Q2:**
> We observe a non-trivial mixture of Rocq tactics appearing inside Lean proofs. In particular, we consistently see destruct-style usage: the model generates Rocq’s destruct tactic in a Lean cases-style syntax that is syntactically mismatched but semantically aligned.
> We also observe cross-syntax references, where Rocq tactics are rendered using Lean conventions. This suggests that the steering vectors encode proof assistant agnostic reasoning strategies. We will add qualitative examples of Rocq-style proofs in the revision.
>
> **W3 & Q1:**
> We performed experiments across different random seeds of sampled data and observed highly consistent performance across steering vectors. We also ablated with randomized vectors. These were created using (i) prompt-pair mismatch and (ii) adversarial prompts taken from the Alpaca dataset [1].
> We find that randomized vectors cause a significant performance drop. Qualitatively, when Alpaca prompts are used, InternLM-2.5 often fails to comment correctly inside proofs, leading to trailing proof errors.
>
> ### Performance Comparison
>
> | Method                                                  | Performance |
> | ------------------------------------------------------- | ----------- |
> | InternLM-2.5 StepProver                                 | 48.2%       |
> | InternLM-2.5 + steering vector 1                        | 66.4%       |
> | InternLM-2.5 + steering vector 2                        | 66.4%       |
> | InternLM-2.5 + steering vector 3                        | 66.8%       |
> | InternLM-2.5 + randomized vector (prompt pair mismatch) | 36.92%      |
> | InternLM-2.5 + randomized vector (Alpaca prompts)       | 19.43%      |
> | CoT                                                     | 44.6%       |
> | Few Shot                                                | 49.18%      |
>
> ### Domain Specific Evaluation
>
> | Prompt Pairs                   | Algebra (50) | Number Theory (50) |
> | ------------------------------ | ------------ | ------------------ |
> | 250 Algebra problems           | 29           | 27                 |
> | 250 Number Theory problems     | 25           | 30                 |
> | InternLM-2.5                   | 25           | 25                 |
> | InternLM-2.5 + steering vector | 30           | 31                 |
>
> These evaluations are on a held out 50 problem set from miniF2F.
>
>
> **W4:**
> We conduct a detailed evaluation of tactic usage differences between proofs generated with and without activation steering.
>
> ### Tactic Usage Analysis
>
> | Tactic   | Without Steering | With Steering |
> | -------- | ---------------- | ------------- |
> | by       | 10.8487          | 9.2773        |
> | linarith | 3.7815           | 3.9244        |
> | rw       | 2.8067           | 2.4118        |
> | norm_num | 2.0924           | 1.8571        |
>
> These are the average tactics used per problem.
>
>
> **Q3:**
> Thank you for pointing out typos in the paper, we will ensure consistency in the revision.
>
> [1] [https://github.com/tatsu-lab/stanford_alpaca](https://github.com/tatsu-lab/stanford_alpaca)

---

> > ### Comment · Reviewer_hgza · 2025-11-28
> >
> > Thanks for the replies! One detail: in the tactic usage analysis you shared, you included `by`, which I think is more the beginning of a tactic-style proof rather than a tactic people actually use as a proving step.
> >
> > Overall, it’s great to see the additional explanations. To be honest, I’m still not fully convinced, after reading the paper, that the steering approach would be beneficial or scalable for provers, but I admit that probably has sth to do with my subjective taste, and I’d prefer not to rely too much on my personal views when evaluating the work. So in short I’ll keep my positive score, but I would encourage the authors to think more about, motivate, and possibly experiment further with whether steering could truly be a viable direction for theorem proving.

---

### Official Review · Reviewer_YYwB · 2025-11-02

**Soundness:** 2
**Presentation:** 3
**Contribution:** 2
**Rating:** 4
**Confidence:** 3

**Summary:**

The paper explores activation steering, a fine-tuning-free, inference-time intervention, to improve LLMs' ability to generate formal mathematical proofs. The authors observe that informal mathematical reasoning expressed in natural language can provide important structural guidance for proof construction, but existing models rarely use it effectively. The authors hypothesize that informal reasoning induces distinct activation patterns in the model’s residual stream, and these can be linearly isolated and reapplied to improve formal proof generation. Concretely, they construct steering vectors that capture these patterns by contrasting model activations between prompts with and without informal reasoning. Injecting these vectors into transformer layers during inference steers the model toward reasoning-rich proof trajectories without changing model weights. The authors evaluate on MiniF2F and PutnamBench and show the proposed method improves theorem-proving success rates across multiple models such as Llemma-7B, InternLM-2, and InternLM2.5-StepProver.

**Strengths:**

1. The concept of steering model activation to improve theorem proving is novel and interesting.
2. The proposed activation steering method is lightweight, fine-tuning-free, requires only forward passes to compute steering vectors, making it readily pluggable into existing LLMs.
3. The authors demonstrate strong gain brought by activation steering on MiniF2F, +18.2%.

**Weaknesses:**

1. With activation steering, while short proofs improve significantly, long or highly compositional proofs show limited benefit and sometimes even degraded performance due to noisy reasoning insertions.
2. The gain is noticeable for InternLM2.5 but modest for smaller models.
3. The robustness of activation steering is unclear with respect to prompts and hyperparameters.
4. It would nice to evaluate the effect of activation steering on more recent state-of-the-art theorem-proving models such as Goedel Prover.
5. The paper lacks comparisons with frontier theorem-proving frameworks such as Seed-Prover [1], Goedel Prover [2], and LLM-based provers such as GPT-5, Qwen-235B, Claude Sonnet, and Grok.



[1] Chen, Luoxin et al. “Seed-Prover: Deep and Broad Reasoning for Automated Theorem Proving.” ArXiv abs/2507.23726 (2025): n. pag.

[2] Lin, Yong et al. “Goedel-Prover-V2: Scaling Formal Theorem Proving with Scaffolded Data Synthesis and Self-Correction.” ArXiv abs/2508.03613 (2025): n. pag.

**Questions:**

Do the authors have any insights on why activation steering brings significant gains on MiniF2F but only minimal improvements on PutnamBench? Similarly, why is the gain more noticeable for InternLM2.5 compared to smaller models? Under what conditions does activation steering tend to work well, and when does it fail?

---

> ### Author Response · Authors · 2025-11-22
>
> We thank the reviewer for recognizing the strengths of the work. We address each concern below:
>
> **W1:**
> We acknowledge the length biases in the proofs documented in Table 3 and Section 5.3. Our analysis shows that steering vectors encode inductive biases from the initial distribution of prompt pairs. These pairs predominantly contain proof steps, leading the instructions to emphasize writing steps rather than constructing longer proofs. For longer proofs, injecting the steering vector at each step can introduce noise.
> However, we view this as an actionable insight rather than a fundamental limitation. This insight also reveals *when and why* informal reasoning helps or hinders formal theorem proving—an understanding that fine-tuning alone cannot provide.
>
> **W2:**
> The differential gains across model scales reflect meaningful differences in internal representations. InternLM2.5 was trained via expert interaction on large-scale synthetic Lean problems, which likely produced a cleaner separation between formal and informal concepts. Prior work (e.g., arXiv:2412.17626) shows that feature formation and isolation is a gradual process. Our hypothesis is that with a larger training corpus, the features captured by InternLM2.5 are more refined than in other models.
>
> **W3:**
> To provide a more robust analysis, we experimented with randomized prompt pairs (matching (p_1) with (p+k)) and two additional data splits. The results are included below.
>
> **Table: Performance on miniF2F**
>
> | Method                                                | Performance |
> | ----------------------------------------------------- | ----------- |
> | InternLM2.5-StepProver                                | 48.2%       |
> | InternLM2.5 + steering vector-1                       | 66.4%       |
> | InternLM2.5 + steering vector-2                       | 66.4%       |
> | InternLM2.5 + steering vector-3                       | 66.8%       |
> | InternLM2.5 + randomized steering vector (mismatched) | 17.37%      |
>
> **W4 and W5:**
> The mentioned models would be methodologically inappropriate for our work as they used for whole proof generation. The scope of our work focuses on step by step theorem proving methods and the activation steering operates at a step level. Applying activation steering at a whole proof generation level would require fundamentally different a much broader concepts to capture.
>
> **Q1:**
> PutnamBench contains graduate level competition problems which are harder than high school olympiad problems in MiniF2F. Secondly, PutnamBench spans advanced topics (real analysis, abstract algebra, topology) making the theorem proving task more harder.

---

### Comment · Area_Chair_2qYe · 2025-11-28
**Respond to the authors’ rebuttal**

Dear Reviewers,

As the discussion phase is nearing its end, please read and respond to the authors’ rebuttal, particularly if it addresses your concerns. Your response is valuable for the final decision.

---

### Meta-Review · Area_Chair_XS3f · 2026-01-07

**Summary:**

- **YYwB** (4): Limited benefits for long/compositional proofs, modest gains on smaller models, unclear robustness to prompts/hyperparameters, missing comparisons with recent SOTA (Goedel Prover, Seed-Prover).
- **hgza** (6): Sensitivity to prompt pairing schemes and dataset domains, modest PutnamBench improvements, ambiguous mechanism vs. style capture, insufficient automatic metrics for "structured reasoning."
- **PYvo** (4): No discussion of dominant long-CoT provers (DeepSeek-Prover-V2, etc.), limiting practical significance; unclear applicability to modern models; missing promised "have tactic" analysis.
- **JcFZ** (0): Minimal novelty beyond existing activation steering (CAA); misconfigured baseline (48.2% vs. reported 65.9%); fragile evaluations lacking statistical significance; no SOTA comparisons.

**Reviewer Concerns:**

Addressed: Robustness partially mitigated via multiple steering vectors and randomized controls; baseline discrepancy acknowledged.

Outstanding: (1) Relevance to long-CoT paradigms remains unclear; (2) Statistical significance tests and comprehensive tactic analysis absent; (3) Cross-system transfer claims lack sufficient evidence; (4) Core novelty question unresolved.

**Reviewer Scores:**

No reviewers seemed to would have changed their score according to the discussion.

---

### Decision · Program_Chairs · 2026-01-26

Reject